# Investigating the Degradation of Mycenaean Glass Artifacts Using Scientific Methods

Maria Kaparou [1,*], Artemios Oikonomou [1,2,3] and Andreas Germanos Karydas [1]

1. Institute of Nuclear and Particle Physics, N.C.S.R. "Demokritos", Ag. Paraskevi, 15341 Athens, Greece; a.oikonomou@inp.demokritos.gr (A.O.); karydas@inp.demokritos.gr (A.G.K.)
2. Department of Conservation of Antiquities and Works of Art, University of West Attica, 12243 Egaleo, Greece
3. Science and Technology in Archaeology and Culture Research Center (STARC), The Cyprus Institute, 2121 Nicosia, Cyprus
* Correspondence: kaparou@inp.demokritos.gr

**Abstract:** Mycenaean vitreous artifacts, such as beads and relief plaques, are highly susceptible to degradation, which can significantly alter their visual characteristics and pose challenges to their taxonomy. The visual manifestation of corrosion on vitreous artifacts, especially glass and faience, has often led to their misclassification, which, in turn, has a significant impact on their interpretation by researchers, often resulting in misleading notions. The present paper constitutes part of an overall study, implemented within the framework of the project, *Myc-MVP: Mycenaean Vitreous Production, A Novel Interdisciplinary Approach Towards Resolving Critical Taxonomy Issues*, which has employed a combination of established, state-of-the-art scientific methods to analyze and identify the specific compositional changes occurring at different spatial dimensions within surface layers, with the overarching aim of contributing to our understanding of the degradation mechanisms of vitreous artifacts and the relevant implications for the archaeological record. Importantly, these findings will yield useful data in devising strategies for the proper classification, management, and preservation of vitreous artifacts in the future. The present study focuses on investigating the relationship between the compositional changes in a subset of 12 (of the overall 126 objects entailed in the project) corroded vitreous artifacts from Mycenaean contexts in the Aegean and the way these are manifested visually, with the application of X-ray fluorescence and LED microscopy. We aim to decipher the nature of corroded objects with the aid of focused analysis. This study delves into degradation processes in glass artifacts, highlighting preservation variations and environmental influences like burial. Coloration, attributed to copper and cobalt oxide, shows some correlation with preservation quality. These chromophore agents potentially induce thermal stresses and corrosion. The complex interplay between chemical composition, environmental conditions, and preservation status underscores the need for comprehensive research. Analyzing the full artifact set using complementary techniques promises deeper insights for secure material classification and cultural heritage preservation.

**Keywords:** Mycenaean vitreous materials; glass; pXRF; glass corrosion; vitreous material classification

## 1. Introduction: Degradation of Archaeological Glasses/Glazes

Glass corrosion and its mechanisms have been studied extensively [1–3] and this is out of the scope of this paper. The main aim of this paper is to offer strategies to tell apart degraded vitreous materials by characterizing glass and understanding its corrosion products with the application of non-invasive spectroscopic techniques, in particular X-ray fluorescence. This paper lies in the idea that Mycenaean vitreous materials, such as faience and glass, suffer a lot from degradation, making it difficult for the non-expert to distinguish them in the field and in museums/storage rooms. It builds upon the selection of objects whose preservation state allows for their secure attribution to a specific material and their equivalents from the same contexts, whose nature is unclear. In this paper, we present the

preliminary results obtained by LED microscopy and micro-XRF applied on a subset of 12 objects of the overall assemblage of vitreous artifacts (126 artifacts).

Archaeological glasses and glazes exhibit intricate networks, typically characterized by silica as the primary network former, accompanied by minor metallic oxides with a high bond strength [1,2]. Ancient glassmakers would incorporate modifiers to the network, in order to lower the silica melting point, and these modifiers are ionically bonded into the glass network. Therefore, alkalis, such as sodium and potassium, function as fluxes, reducing glass viscosity, while alkaline earth metals like calcium and magnesium in controlled amounts enhance glass durability and deter recrystallization. The type and amount of these modifiers, alongside each major component and the overall composition of the glass within the glass network, determine the decay rate and overall degradation of glass artifacts, which is also dependent on environmental factors.

Late Bronze Age glass and, thus, Mycenaean glass, is typically plant ash-based, with sodium, potassium, and magnesium oxides being particularly important in a glass network [4]. The silica-to-alkali ratio plays a crucial role in the durability of the glassy layer; higher concentrations of alkali in wet environments are more prone to leaching, making the glass more susceptible to decay. Studies indicate that when glass has in its structure below 40 wt% (66 mol%) silica, it is more likely to corrode, and recent research suggests this threshold is around 37 wt% (62 mol%) [2]. Hence, glasses with lower silica concentrations tend to degrade more. Primarily, it is the alkali that undergoes decay in glasses, especially in the presence of very low concentrations of calcium or magnesium, which act as stabilizers of the network. Upon initial corrosion, the alkalis are typically the first to be lost from the glass surface, through the process of leaching. Notably, archaeological glasses, mainly fluxed with sodium or potassium, show that those containing the smaller cation, sodium, are more stable than those with high potassium concentrations, as sodium tightly binds within the silica network [5]. The higher mobility of potassium ions in comparison to sodium ions leads to a dissolution rate for potassium-rich buried glass about one order of magnitude higher than that of sodium-rich glasses [6]. Therefore, Mycenaean glasses, having as an alkali flux plant ashes and, thus, higher concentrations of potassium, as well as given their small size, are particularly vulnerable [2].

While the presence of calcium, and to a lesser extent magnesium, is likely to enhance glass durability, the calcium-to-alkali proportion is significant [2]. A greater proportion of calcium, ideally around 2.5–4 wt% (6–10 mol%), to sodium or potassium results in a more stable glass. At these concentrations, it binds alkali into the glass matrix, reducing alkali leaching and forming a more stable "gel" surface. However, excessive calcium, above 6 wt% (15 mol%), leads to corrosion, as alkalis and calcium leach out, breaking the silica network and forming an unstable silica layer, especially at a pH below 3 [7]. In all these instances, corrosion produces a silica-rich layer on the glass surface, especially when the bulk composition has low silica content [2]. Overall, as stated earlier, Mycenaean glass, due to its plant ash-based composition and, thus, its occurrence of variable potassium and calcium content, is more susceptible to degradation, usually developing a thick hydration layer which also depends on the burial environment. Often, the weathering crust exhibits a laminar structure, featuring individual parallel layers with varying thicknesses, ranging from less than 1 μm to approximately 25 μm. The deterioration process unfolds cyclically, giving rise to discernible layers within the weathering crust.

In certain instances, alkali-deficient layers serve as a protective barrier for the remaining glass, impeding water access and thereby decelerating the formation of new layers. Whether the crust provides protection depends primarily on the glass composition, as well as the pH and temperature of the leaching solution. There are cases in which the layers cannot act protectively, thus allowing the weathering process to proceed inwards into the bulk [1]. Alkali environments pose a great threat, as they attack and break down the molecular framework of the glass, the silica network, ultimately leading to the complete dissolution of the glass. When elevated temperatures occur, the deterioration process is accelerated [8]. Variances in the burial environment, particularly localized differences, can

result in variations in the extent and appearance of glass weathering within the same site, sometimes even within a single piece of glass [9].

The most common visual manifestations of degradation on archaeological glass include a total loss of their glassy nature, dulling, iridescence, weeping, opaque or opalescent weathering, pitting and hollowing, cracking of the surface, discoloration, concentric layers and indications of glass dissolution/recrystallisation patterns, a chalk-like or bone-like texture, and crizzling. These manifestations were taken into account in the course of this research.

In particular, complete *loss of their glassy nature* occurs when glass is severely deteriorated, existing only as a mass of silica gel that can be challenging to identify as glass, resulting in mismanagement in their archaeological classification and also the conservation strategy applied. The thickness of the weathering crust varies based on the chemical stability of the glass and the burial conditions, while corrosion products may entirely replace the original glass, which is a very common phenomenon when it comes to Mycenaean glass artifacts. *Dulling* refers to the loss of the original clarity and transparency of glass, resulting from changes in the surface composition that alter the refractive index. Under neutral or acidic conditions, alkalis are commonly leached out from the initial layers of the glass onto the surface. When interacting with humidity and moisture in the environment, they have a tendency to create corrosion products, such as salts, which accumulate on the object's surface, resulting in the loss of the original clarity [3]. *Iridescence* refers to weathering crusts of numerous thin alternating layers of air and weathered glass forming, due to the loss of alkalis, and leading to interference between reflected light rays. *Weeping* as a term is used by archaeologists to describe the formation of "tears" or "sweat" on the surface of excavated iron due to the hygroscopic nature of iron chloride salts [10]. In characterizing glass degradation phenomena, weeping refers to the presence of the distinct moist films or droplets on the surface of the deteriorated object. It can also be found as "sweating". *Opaque weathering* is marked by opaque, usually white, areas on the surface gradually penetrating deeper into the glass. It has a laminar structure, which may be tightly bound and extend throughout the surface, while it can also be superficial. The initial phase is sometimes termed *milky weathering*, due to small white spots or streaks, whereas, when degradation has advanced, colors may range from black or brown to a mottled polychrome. At its most extreme, it is described as *enamel-like weathering*, presenting as a thick, variably colored covering. *Pitting* can occur when deterioration starts at a specific point, either on or just below the surface, potentially creating concentric circles. When weathering is lost, it leaves a hole or pit in the otherwise undamaged glass. *Concentric layers* are usually associated with pitting as they tend to form around the holes or pits. They refer to a complicated succession of concentric and radial features (laminae and lamination) creating an intense network of cracks which can resemble a crater. *Cracking* can affect the visual appearance and stability of glass, due to the shrinkage of the alkali-deficient layer, as a result of temperature and humidity changes. *Discoloration* of glass occurs due to the migration or alteration of coloring ions and other trace elements. These ions may leach out of the glass or be absorbed from the environment. Iron and manganese contribute to the blackening of weathering crusts, while contact with copper corrosion products can induce green staining. Certain ions, notably manganese and copper, may change color through oxidation. *Crizzling* has been identified as the major alteration symptom for glass objects stored in museums and in private collections, referring to diminished transparency owing to very fine surface crazing, exhibited as minute cracks.

The development of either of these manifestations is influenced by both the physicochemical properties of glass and the environmental conditions that developed during its burial. A single glass object may exhibit more than one form of alteration and even multiple ones.

## 2. Materials and Methods

This paper is part of the project entitled, *Myc-MVP: Mycenaean Vitreous Production, A Novel Interdisciplinary Approach Towards Resolving Critical Taxonomy Issues*, funded by

the Hellenic Foundation for Research and Innovation, and is hosted at the Institute of Nuclear and Particle Physics, NCSR Demokritos, in Athens. The objects studied, in total 126 vitreous objects, come from various Mycenaean contexts and are divided in two main categories: glass and faience. Vitreous materials exhibiting corrosion of various levels were selected, with care being taken to sample from contexts in which samples of an ambiguous nature had their "healthy" equivalent in terms of typology. For the purposes of this paper, 12 objects were selected, and in particular, MVP-2187_14a,b and MVP-2189_15a,b were retrieved from the Mycenaean Cemetery in Spata, Attica; MVP-6592_56 a,b,c in Prosymna, Peloponnese; and MVP-10586_71a,b,c,d,e from a burial context in ancient Asine in the Peloponnese (Figure 1). The objects under study were divided into five groups based on their typological criteria and different degrees of degradation and further investigated with an LED microscope and micro-XRF (Figure 2). The samples date from LBAIIIA to LBAIIIB (14th to 13th BCE) and comprise exclusively relief plaques. This is not random, since the Mycenaean glass industry stands out distinctly from similar industries in the Mediterranean and the Near East of the times. It is characterized by its predominant focus on crafting glass jewelry adornments, rather than producing larger sized objects, like vessels [4].

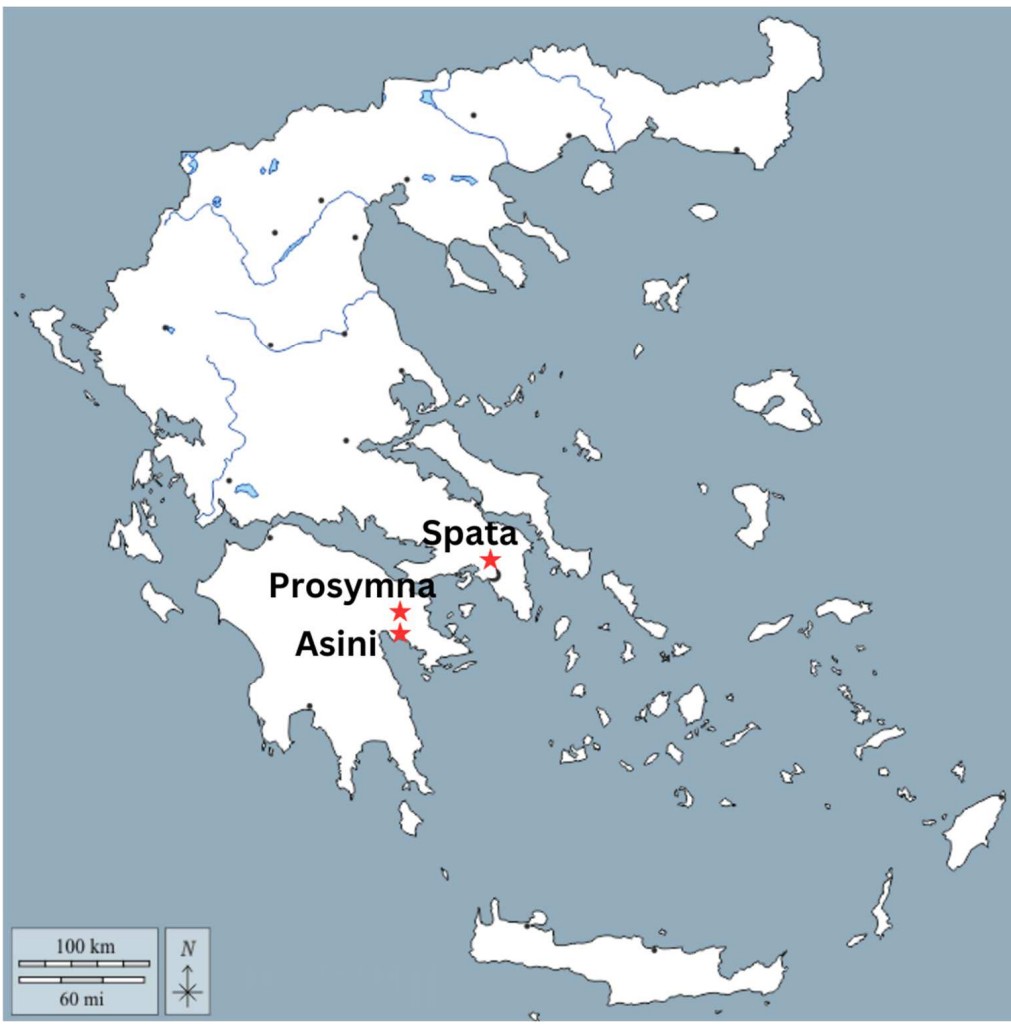

**Figure 1.** Map of Greece exhibiting the areas of origin for the Mycenaean samples under study.

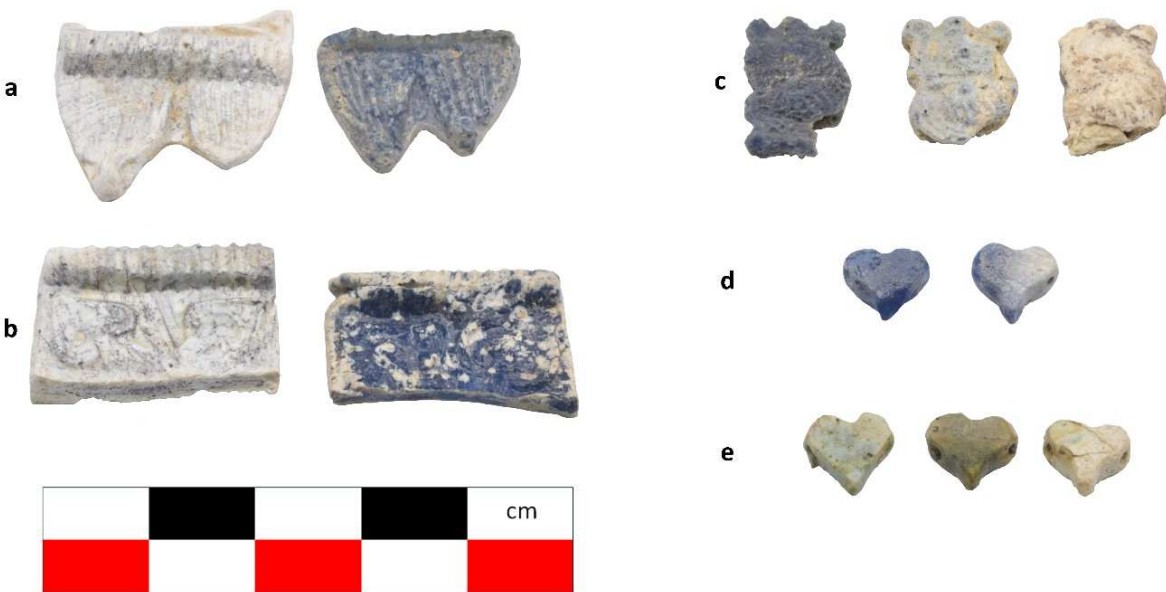

**Figure 2.** The samples exhibit different preservation states presented from the best preserved on the left. (**a**) Mycenaean relief plaques (sample numbers: MVP-2187_14a,b) from Spata, Attica; flat back; triangular shapes representing ivy leaves in a row with 'thysanos' at the bottom; straight sides; one hole pierced through on top. (**b**) Mycenaean relief plaques (sample numbers: MVP-2189_15a,b) from Spata, Attica; flat back; ivy leaf decoration; straight sides; one hole pierced through on top. (**c**) Mycenaean relief plaques (sample numbers: MVP-6592_56a,b,c) from Prosymna, Peloponnese; adjacent nautilus shapes; flat back; two holes pierced through top. (**d**,**e**) Ivy leaf shape; flat back; two holes pierced through top and side of leaf on each side, joining at an angle (sample numbers: MVP-10586_71a,b,c,d,e) from ancient Asine in the Peloponnese.

## 3. Micro-XRF

*The INPP Portable Micro-XRF Spectrometer*

The micro-XRF analysis technique allows for rapid, non-destructive, and sensitive elemental analysis of major, minor, and trace elements in archaeological and historical artifacts. The microprobe XRF analysis was carried out using a customized version of the so-called portable Artax spectrometer (Bruker Nano GmbH, Berlin, Germany) at the X-ray Fluorescence Laboratory of the Institute of Nuclear and Particle Physics (INPP) at the NCSR "Demokritos," Athens. The INPP micro-XRF spectrometer is composed of an X-ray microfocus Rh anode tube (spot size $50 \times 50$ μm, maximum high voltage 50 kV, tube current 0.6 mA, 30 W maximum power consumption, beryllium window 0.2 mm thickness) and a polycapillary X-ray lens as a focusing optical element. The X-ray lens provides spatial resolution (spot size) for the filtered exciting beam at the order of ~50 μm CuK$\alpha$ when the exciting beam is filtered [11]. The X-ray detection chain also consists of a thermoelectrically cooled 10 mm$^2$ silicon drift detector with 146 eV FWHM MnK$\alpha$ and 10 kcps. A color CCD camera (with approximately $\times 13$ times magnification) combined with a dimmable white LED and a spot laser beam assures the reproducible positioning of the measuring probe, as well as visualization and documentation of the analyzed area. Three stepping motors coupled with the spectrometer head allow three-dimensional movement for elemental mapping and precise setting of the analysis spot at the focal distance of the polycapillary lens.

The objects were analyzed in a scanning mode of an area approximately $0.5 \times 0.3$ mm, and 18 measurements were performed on each sample at a step of 0.1 mm per measurement. The acquisition time was set at 50 s per measurement. The voltage and the current were set at 50 kV and 600 μA, respectively, while the measurements were performed in the no filter mode. Calibration and precision were made using BAMS005 and Corning A standard

reference glasses (Table 1). The spectra were processed with the open software package Pymca for the quantification results [11,12].

**Table 1.** Oxide concentrations in ppm (otherwise stated) determined by means of micro-XRF analysis of the reference glasses BAM S005 and Corning A at 50 kV. Precision (%) is reported. Certified data for Corning A were derived from Adlington, 2017 [13]. LoD: limit of detection.

| Oxide (ppm) | BAMS005 (Certified) | BAMS005 Measured (n = 6) | Difference (Rel. %) | Precision (%) |
|---|---|---|---|---|
| $SiO_2$ (%) | 71 | 70.9 | −0.1 | 0.09 |
| $K_2O$ (%) | 0.7 | 0.7 | −6.3 | 0.01 |
| CaO (%) | 10.5 | 9.4 | −10.5 | 0.2 |
| $TiO_2$ | 164 | 108 | −34.4 | 8 |
| $MnO_2$ | 124 | 150 | 21.3 | 5 |
| $Fe_2O_3$ | 422 | 426 | 0.9 | 7 |
| CoO | 49 | 55 | 11.5 | 2 |
| NiO | 59 | 60 | 2.1 | 2 |
| CuO | 112 | 117 | 4.8 | 2 |
| ZnO | 203 | 230 | 13.3 | 6 |
| $As_2O_3$ | 132 | 128 | −3.1 | 6 |
| SrO | 151 | 181 | 19.8 | 6 |
| $ZrO_2$ | 842 | 983 | 16.8 | 38 |
| $SnO_2$ | <LoD | 296 | | |
| $Sb_2O_5$ | <LoD | 274 | | |
| BaO | 115 | 150 | 30.4 | 13 |
| PbO | 202 | 242 | 19.9 | 9 |
| Oxide (%) | Corning A (Certified) | Corning A (Measured) | Difference (Rel. %) | |
| $SiO_2$ | 66.56 | 66.9 | −0.4 | |
| $K_2O$ | 2.87 | 2.75 | 4.3 | |
| CaO | 5.03 | 5.2 | −3.6 | |
| $TiO_2$ | 0.79 | 0.83 | −4.4 | |
| $MnO_2$ | 1.00 | 0.99 | 1.0 | |
| $Fe_2O_3$ | 1.09 | 1.05 | 4.2 | |
| CoO | 0.17 | 0.16 | 6.3 | |
| NiO | 0.02 | 0.02 | 17.5 | |
| CuO | 1.17 | 1.09 | 7.4 | |
| ZnO | 0.04 | 0.04 | −7.6 | |
| SrO | 0.1 | 0.10 | −3.0 | |
| $ZrO_2$ | 0.01 | 0.01 | 11.0 | |
| $SnO_2$ | 0.19 | 0.18 | 3.6 | |
| $Sb_2O_5$ | 1.75 | 1.34 | 30.2 | |
| BaO | 0.46 | 0.39 | 18.5 | |
| PbO | 0.07 | 0.06 | 12.1 | |

The objects were investigated under an LED Dino-Lite microscope, in order to make a first assessment of the corrosion products and identify areas for the subsequent micro-XRF analysis (Table 2). It utilizes advanced LED lighting technology to provide bright, uniform illumination for microscopic examination, providing a feature-rich solution for microscopic inspection at up to 900× magnification and 5 megapixel resolution, ideal for this superficial study and first classification of the artifacts.

**Table 2.** Overall table showing the objects under study including microscopic and macroscopic observations.

| Group | Object | Material | Observations | Corrosion Patterns | Micro-XRF Analysis Area | Microscopic Photo |
|---|---|---|---|---|---|---|
| 1 | MVP-2187_14b  | Glass | Condition: Heavily corroded. Partial loss of glassy surface/texture. | Pitting, crustration, milky, enamel-like surface, discoloration. |  |  |
| | MVP-2187_14a  | Unclear | Condition: Heavily corroded. Excessive corrosion at the decorative elements, no flaking off. | Pitting, crustration, discoloration, iridescence, enamel-like surface, microcracking. |  |  |
| 2 | MVP-2189_15b  | Unclear | Condition: Medium–heavily corroded. Excessive corrosion at the decorative elements. | Crustration, discoloration, pitting. |  |  |
| | MVP-2189_15a  | Glass | Condition: Medium–heavily corroded. Excessive corrosion at the decorative elements. | Pitting, crustration, metal oxidation. |  |  |

**Table 2.** *Cont.*

| Group | Object | Material | Observations | Corrosion Patterns | Micro-XRF Analysis Area | Microscopic Photo |
|---|---|---|---|---|---|---|
| 3 | MVP-6592_56a  | Glass | Condition: Heavily corroded. Fragile object having poor coherence. | Pitting, crustration. |  |  |
| | MVP-6592_56b  | Unclear | Condition: Heavily corroded. Fragile object having poor coherence, ridges. | Pitting, crustration, iridescence, discoloration. |  |  |
| | MVP-6592_56c  | Unclear | Condition: Heavily corroded. Fragile object having poor coherence, ridges. | Pitting, crustration, iridescence. |  |  |
| 4 | MVP-10586_71a  | Glass | Condition: Well–medium corroded. Good coherence. | Pitting, discoloration, crustration. |  |  |
| | MVP-10586_71b  | Glass | Condition: Well–medium corroded. Air bubbles, two-colored glass, good coherence. | Pitting, discoloration, crustration. |  |  |

| Group | Object | Material | Observations | Corrosion Patterns | Micro-XRF Analysis Area | Microscopic Photo |
|-------|--------|----------|--------------|--------------------|------------------------|-------------------|
| 5 | MVP-10586_71c  | Glass | Condition: Well–medium corroded. Good coherence. | Pitting, discoloration, metal oxidation, crustration. |  |  |
| | MVP-10586_71d  | Unclear | Condition: Well–medium corroded. Brownish hue, air bubbles, good coherence. | Pitting, discoloration, metal oxidation, crustration. |  |  |
| | MVP-10586_71e  | Unclear | Condition: Well–medium corroded. Good coherence. | Pitting, discoloration, metal oxidation, crustration, microcracking. |  |  |

## 4. Results

### 4.1. LED Microscopy

All samples exhibit different degrees of alterations, which in some cases obscure the real nature of the artifact. It should be taken into account that soda–lime–silica glass exhibits distinct characteristics when it comes to corrosion. Specifically, when an artifact is exposed to acidic soil, one can observe the formation of both isolated and interconnected fissures. Conversely, in neutral and alkaline soil environments, there is an uptick in the occurrence and depth of pits. The rate of pit formation correlates with the presence of alkaline oxides in the glass composition. Furthermore, there is a notable escalation in the diffusion of surface degradation under alkaline conditions [3].

In MVP14b, the glassy layer was not totally lost, also preserving a clear indication of the initial color, whereas in MVP14a, the created milky-like surface as a result of severe degradation and the total discoloration would make its safe classification difficult for the non-expert. Of the second set of samples (MVP15a,b), MVP15b, while discolored and with obvious corrosion products exhibited in the formation of local crust, pitting, and micro-pitting, probably indicative of its burial in a neutral or alkaline environment, the obvious loss of its glassy layer, and its opaque milky-like surface, is still quite easily classifiable as glass. MVP15a, on the other hand, is totally discolored and an opaque milky-like surface has replaced the initial glass.

The MVP56a,b,c plaques are characteristic of how different stages of corrosion are depicted in glass. The glassy layer is not obvious in any of the objects; nevertheless, MVP56a is obviously a glass object, while this is not clear in MVP56c, which has a bone-like appearance and is quite fragile, with a total loss of its initial color and no areas of pristine glass detected. In artifacts MVP71a,b,c,d,e, a glassy layer is still present. Nevertheless, degradation has influenced the samples in different ways, despite the fact that they derive from the same context both in terms of locality and chronology. There are different degrees of discoloration of the samples, with sample MVP71d having turned into brown, whereas MVP71e is totally white. The glassy layer is obvious in samples MVP71a,b, while its detection macroscopically is not feasible in the other samples. MVP71c has suffered discoloration and there is an obvious formation of a gel- like, milky-like layer.

### 4.2. Micro-XRF Results

The data acquired by the micro-XRF exhibit interesting patterns which can lead to useful conclusions regarding the mechanism of corrosion of the samples.

The relative intensities were calculated by taking Corning A as a reference sample. In particular, the composition of Corning A reflects a plant ash composition which was considered as the baseline. Therefore, the counts per second (cps) for each peak of the acquired spectra were divided by the cps of Corning A, giving us an idea of how the values of the specific oxides were altered in comparison to those of the normal, uncorroded, referenced glass Corning A. In the first approximation, values above 1 meant that there was an excess of a specific oxide, while values below 1 signified the opposite (Figure 3) [11].

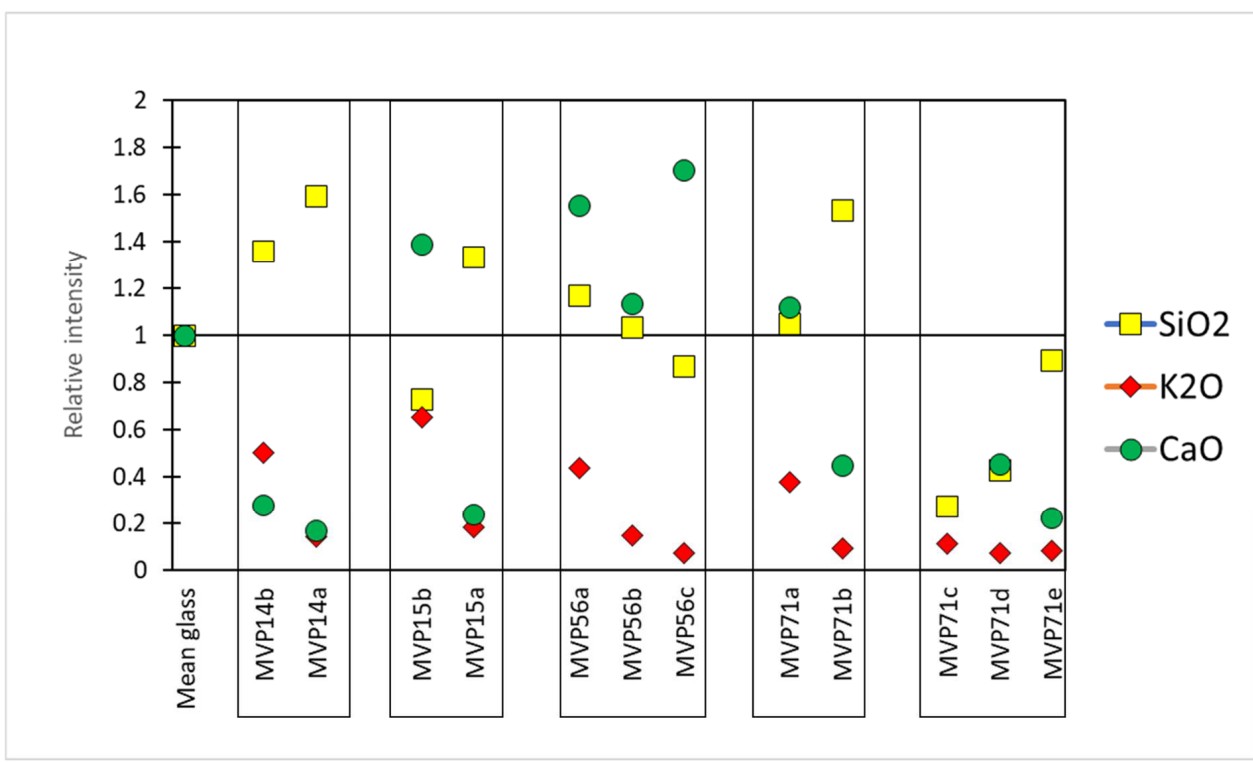

**Figure 3.** The relative intensities of the basic glass components $SiO_2$, $K_2O$, and CaO compared to the Corning A reference glass (Mean glass). The relative intensity for CaO for object MVP71c was 4.8 (not shown on the graph).

In set MVP14a,b, object MVP14b preserved a clear glassy layer with potassium being present, although somewhat leached. Its less preserved equivalent, MVP14a exhibits higher levels of silica and has low levels of potassium and calcium oxides. This can be macroscopically seen in the occurrence of pitting and micro-pitting, which results from

the silica network dissolution, the clear discoloration of the artifact, and the formation of a gel-like layer.

Similar phenomena are seen in glasses MVP15a,b. In particular, MVP15a exhibits some enrichment in silica and leaching in alkali and lime. This resulted in the formation of a local crust, and there is severe discoloration and initiation of a gel-like layer/opaque, milky-like surface, typical of degraded glass beads. However, in MVP15b, which is better preserved macroscopically, the potash levels are low, yet close to what would be expected for plant ash glasses and its lime levels are also quite normal (7 wt%), falling within the expected range for soda–lime–silica glasses, while there is some depletion in silica, which is related to the formation of micro-pitting [3].

The relief plaques MVP56a,b,c exhibit three visually different stages of corrosion In the same type of object. According to the analytical data, the silica network gradually broke down and potassium was leached in correspondence with the visual characteristics of the objects and the level of their corrosion.

Sample MVP71a preserves a glassy layer with expected levels of silica and calcium oxide. Nevertheless, depletion in potassium seems to have started and this is also exhibited in the initiation of discoloration at patches. Its less preserved equivalent, namely MVP71b, exhibits higher levels of silica and is depleted in potassium and calcium oxides. This is macroscopically seen in the clear initiation of discoloration, with some white striking and weeping.

In the next set (MVP71c,d,e), the sample that clearly maintains a glassy layer is MVP71c, despite the obvious degradation. A plausible scenario for its instability could be its really high levels of calcium oxide of the order of 20 wt%. While it has been generally accepted in the literature that calcium oxide benefits glass durability against corrosion [2], the complete demonstration of the beneficial impact of alkaline earths on enhancing glass stability against alteration has not been fully documented. It has been indicated that elevating the CaO to $SiO_2$ ratio enhances glass stability, whereas calcium oxide concentrations exceeding 15 wt% lead to rapid glass instability [14]. Thus, for calcium oxide, while acting as a stabilizer, its excess presence can cause the exact opposite effect, destabilizing the glass network. In the case of sample MVP71c, it is unclear whether its high levels of calcium are an intentional occurrence or a result of glass degradation. Several studies have suggested that calcium ions ($Ca^{2+}$) and magnesium ions ($Mg^{2+}$) exhibit mobility comparable to alkalis within the hydrated layer under atmospheric conditions [15]. Thus, enrichment in calcium oxide could have occurred after unearthing the artifact and after it has been exposed to the atmosphere. Other studies have noted that even for glass samples buried in soil, the predominant issue observed is the development of de-alkalinization layers. A study aiming to monitor the gradual de-alkalinization of a glass bulk composition using laser-induced breakdown spectroscopy (LIBS) reported an augmentation in the intensity signals of calcium and sodium on the surface of the glass [16]. Given the fact that its equivalent artifacts from the same context do not exhibit such high levels of calcium, its elevated contents do not seem to constitute an intentional addition, but rather an accidental occurrence during their making or a different response to environmental factors before and after its unearthing.

Sample MVP71d appears totally discolored and acquired a brown tinge. Darkening is closely associated with the oxidation of specific leached ions, such as iron, manganese, and copper, resulting in a change in the color of the weathering crusts. Additionally, darkening can occur due to the production of hydrogen sulfide by sulfur-reducing bacteria in anaerobic environments, leading to the formation of lead sulfide [3]. The latter scenario only takes place when glass contains a high concentration of lead oxide and is buried under anaerobic conditions. In the present object, only traces of lead were identified. In the case of MVP71d, the presence of copper, manganese, and iron contributed to the darkening of glass, accompanied by the formation of brownish pits. Manganese and iron oxides are present in the glass as impurities, at 0.11 and 0.4 wt%, respectively, and in combination with the high levels of copper oxide (3.5 wt%), are likely to have caused the color alteration. The analysis of MVP71d also revealed leaching of the alkalis and low levels of silica underlying the

breakage of the silica network. The silica network dissolution is exhibited macroscopically with obvious pitting and micro-pitting. On a similar note, MVP71e seems to have almost lost its glassy layer and exhibits almost total discoloration with some patches of the initial color preserved. This sample is depleted in alkali, while silica is present at expected levels. Leaching resulted in the formation of a local crust and gel-like layer, causing pitting, with the surface being anomalous with the formation of corrosion products.

Regarding the chromophores, specific ratios were calculated in order to understand fluctuations in their composition in the cases of the corroded objects. In particular, the composition of the colorant oxides was divided to the mean value of published data from LBA glass which can be considered as the composition in chromophores of a typical LBA glass used herein as a reference glass [17–20]. Therefore, in the first approximation, similarly to the base glass composition (see above), values above 1 meant that there was an excess of this specific chromophore oxide, while values below 1 the opposite (Figure 4) [11].

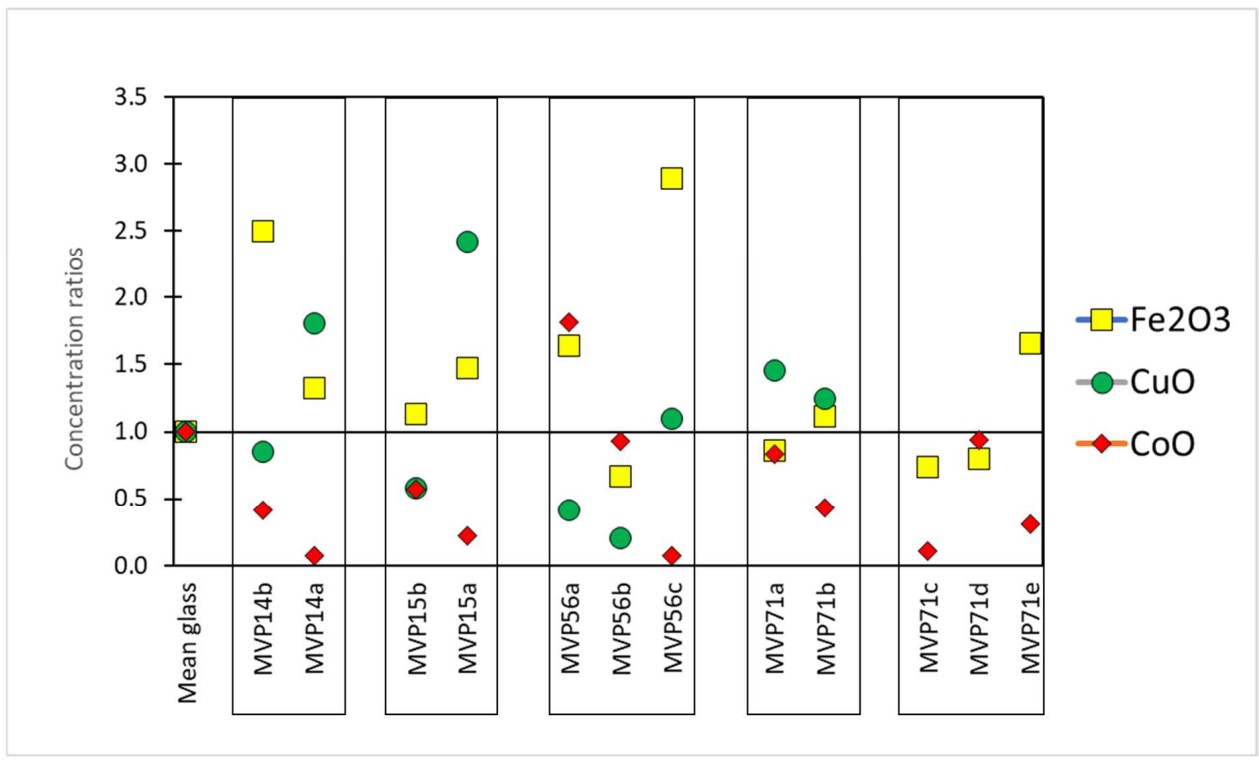

**Figure 4.** The concentration ratios of the colorant components $Fe_2O_3$, CoO, and CuO compared to the average of published Late Bronze Age typical glass, used herein as a reference glass. Concentration ratios for CuO for MVP71c,d,e were 7.9, 20.0, and 8.0, respectively (not shown on the graph). Published data used for this graph [17–20].

The coloration of the samples was almost always due to a combination of copper and cobalt oxide (Table 3). The concentrations in Table 3 represent apparent concentrations of those oxides without considering the complex stratigraphy of the objects' surface. Those results should be considered valid in a semi-quantitative manner. Lighter blue and undiagnostic hues are richer in copper oxide (3198–35,476 ppm CuO) in comparison to the darker hues (726–2571 ppm CuO), which in turn exhibit higher levels of cobalt oxide (417–1821 ppm). The darker blue color being preserved is due to cobalt ions, since their molar extinction coefficient is much higher than that of copper ions. MVP56b, which preserved a light blue tinge, exhibited cobalt oxide levels at 934 ppm and copper oxide levels at 257 ppm, with cobalt thus masking copper as a chromophore. The levels of cobalt and copper oxides follow a declining pattern, in most cases, in correspondence with the preservation state of the object. Therefore, irrespective of what the principal colorant was,

the better preserved objects seemed to retain the coloring agents more as expected, with the exception of MVP56b, which seemed to have been colored only with cobalt oxide. This is probably explained by the fact that the analysis was performed on corroded layers and not on the pristine glass. The relation of the colorants to the process of degradation was not addressed adequately. The thermal stresses induced by the presence of cobalt and copper chromophores and spinel particles have been seen to have been likely responsible for the large cracks filled with corrosion precipitates found in the 19th century green and blue enamels from Barcelona [21]. In general, the behavior of chromophore agents is not fully documented.

**Table 3.** Semi-quantitative estimations of the iron, cobalt, and copper oxides derived by micro-XRF. Σ (%) refers to the statistical precision of the measurements.

| | | $Fe_2O_3$ (ppm) | $\sigma$ (%) | CoO (ppm) | $\sigma$ (%) | CuO (ppm) | $\sigma$ (%) |
|---|---|---|---|---|---|---|---|
| **Group 1** | MVP-2187_14a | 8118 | 0.2 | 73 | 4.3 | 3198 | 0.2 |
| | MVP-2187_14b | 15,269 | 0.1 | 417 | 1.1 | 1513 | 0.3 |
| **Group 2** | MVP-2189_15a | 9024 | 0.1 | 221 | 1.6 | 4272 | 0.2 |
| | MVP-2189_15b | 6904 | 0.2 | 576 | 0.7 | 1023 | 0.4 |
| **Group 3** | MVP-6592_56a | 10,041 | 0.1 | 1821 | 0.3 | 726 | 0.5 |
| | MVP-6592_56b | 4067 | 0.2 | 934 | 0.5 | 357 | 0.7 |
| | MVP-6592_56c | 17,685 | 0.1 | 77 | 4.9 | 1943 | 0.3 |
| **Group 4** | MVP-10586_71a | 5277 | 0.2 | 835 | 0.5 | 2571 | 0.2 |
| | MVP-10586_71b | 6814 | 0.2 | 434 | 0.9 | 2209 | 0.2 |
| **Group 5** | MVP-10586_71c | 4511 | 0.2 | 104 | 3.0 | 13,983 | 0.1 |
| | MVP-10586_71d | 4900 | 0.2 | 947 | 0.6 | 35,476 | 0.1 |
| | MVP-10586_71e | 10,152 | 0.1 | 308 | 1.3 | 14,208 | 0.1 |

## 5. Conclusions

The principal aim of the overall project, Myc-MVP, is to offer a means to the community of accurately classifying corroded vitreous materials of an ambiguous nature, without the aid of a focused analysis. To achieve this, a vast array of samples exhibiting different manifestations of degradation were selected and studied via a focused analysis aspiring to ultimately form a corpus of data that will aid in safe material attribution by using only visual characteristics.

Thus, samples of a clear nature were set against their poorly preserved equivalents. With the aid of a portable USB microscope, different visual manifestations of degradation were seen. MVP14b- 15b- 71a- 71b retain some of their glassy layers and show clear indications of their initial color, making them relatively easy to classify. This is also the case with MVP 56a which, despite the absence of a distinct glassy layer, is identifiable as glass. MVP14a- 15a- 56b- 56c- 71c- 71d- 71e, however, exhibit severe degradation with milky-like surfaces and total discoloration or a bone-like texture, making their safe classification challenging for non-experts.

The current study offers insights into the degradation and alteration processes observed in various glass artifacts, shedding light on the chemical composition and physical manifestations of these changes. Several key conclusions can be drawn from the analysis of the present artifacts. There is a great variability in glass preservation; different glass samples exhibit varying degrees of preservation, with some maintaining a clear glassy layer and others showing signs of degradation such as discoloration, formation of crusts, and gel-like layers, despite belonging to the same context and, thus, leading to the expectation of exhibiting similar behavior. Environmental conditions, such as burial in soil or exposure to atmospheric conditions, can significantly influence glass degradation. While calcium oxide is generally considered to enhance glass durability, excessive levels above 15 wt% can lead to rapid glass instability. The presence of high calcium oxide levels in some samples, such

as MVP71c, may have contributed to their degradation despite calcium oxide's stabilizing potential, indicating a complex interplay between chemical composition and stability.

The coloration of glass artifacts is attributed to the presence of copper and cobalt oxide, with darker hues containing higher levels of cobalt oxide. The preservation state of the objects influences their retention of coloring agents, with better preserved objects exhibiting more consistent coloration. Chromophore agents, such as cobalt and copper ions, have been seen in the literature to be able to induce thermal stresses and contribute to the formation of cracks and corrosion precipitates in glass artifacts. However, the exact mechanisms of their behavior and their relationship to the degradation process require further investigation. It is clear though that there is a clear relation between them.

Overall, this analysis highlights the complex interplay between chemical composition, environmental factors, and preservation state in determining the degradation and alteration of glass artifacts. Further research into the full set of 126 artifacts with the aid of complementary techniques will enhance a better understanding of these processes and aid in developing strategies for the more secure classification of vitreous materials, alongside the preservation and conservation of cultural heritage materials.

**Author Contributions:** Conceptualization, A.O. and M.K.; Methodology, A.O., M.K. and A.G.K.; Formal analysis, A.O. and M.K.; Investigation, A.O. and M.K.; Data curation, A.O. and A.G.K.; Writing—original draft, A.O. and M.K.; Writing—review & editing, A.O., M.K. and A.G.K.; Supervision, A.O. All authors have read and agreed to the published version of the manuscript.

**Funding:** "3rd Call for H.F.R.I. Research Projects to support Post-Doctoral Researchers" (Project Number: 7195).

**Data Availability Statement:** The data is available by the authors upon request.

**Acknowledgments:** This research project was supported by the Hellenic Foundation for Research and Innovation (H.F.R.I.) under the "3rd Call for H.F.R.I. Research Projects to support Post-Doctoral Researchers" (Project Number: 7195). The authors would like to thank Ourania Kordali and Kalliopi Tsampa for the fruitful discussions regarding the corrosion products of the objects under study and important analytical considerations, respectively. The authors would also like to thank them for their overall collaboration within the framework of this project.

**Conflicts of Interest:** The authors declare no conflict of interest.

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
