# Peer review of "Investigating the Degradation of Mycenaean Glass Artifacts Using Scientific Methods"

_heritage, doi:10.3390/heritage7030083_

Round 1
Reviewer 1 Report
Comments and Suggestions for Authors
The work “Investigating the Degradation of Mycenean Glass Artefacts Using Scientific Methods: Preliminary observations” aims to “offer strategies to tell apart degraded vitreous materials through characterizing glass and understanding its corrosion products with the application of non-invasive spectroscopic techniques- in particular X-ray based techniques and brings interesting data towards the relationship between the compositional changes of corroded vitreous artefacts and its visual effects. However, major alterations must be provided in order to guarantee the organization of the data as well as the better understanding of the work.
#1 – The abstract should briefly indicate the main results of the work. Please, add a synthesis of them in this section.
#2 – Page 1 “The main aim of this paper is to offer strategies to tell apart degraded vitreous materials through characterizing glass and understanding its corrosion products with the application of non-invasive spectroscopic techniques- in particular X-ray based techniques. Since the authors perform only XRF to characterize the studied samples, I recommend using “technique” instead of “techniques”.
# 3 – Despite affirming that the paper will not cover glass corrosion and its mechanisms, the authors describe in detail the main visual forms of alteration which occurs in archaeological glass. In my opinion, the description given should be summarized, so the author can explain better what the real issue of the work is. Otherwise, the audience may think the work .
# 4 – Pg 2 – Lines 6-9 “In this paper we present the preliminary results obtained by LED Microscopy and micro-XRF applied on a subset of 12 objects of the overall assemblage of vitreous artefacts (126 artifacts). I suggest moving this information for section Materials and methods.
# 5 – Pg 2 – Lines 41-44 “Therefore, Mycenaean glasses, having as an alkali flux plant ashes and, thus, higher concentrations of potassium and given their small size are particularly vulnerable.”
# 6 – The paper does not present a Material and methods section, although it is a mandatory item for Heritage Manuscript Submissions. What were the techniques employed in this work? What were the conditions of the analysis performed by the authors? The information about the studied samples and the techniques are spread throughout the text and it must be organized in just one section. This description is fundamental to highlight the importance of the work and also to guarantee the reproducibility of the methodology.
# 7 – The section of “results” is where the author presents the data gathered by the work, and not the conditions of the analysis. “Vitreous materials under study” (page 4), the first period of LED Microscopy subsection (page 5), initial paragraphs of Micro-XRF subsection, must be relocated to Materials and Methods section.
# 8 – How can the authors explain the increase or depletion on the levels of chemical elements if only the altered layers were analysed?
# 9 – I suggest the author to use “glassy layer” to describe the effect observed macro and microscopically, since “glassy phase” can induce the audience to think that it is related to chemical alteration identified by other type of analysis.
# 10 – A scale should be included in the images of the samples and its micrography.
# 11 – Page 13, Lines 21-23 “The concentrations in Table 3 represent apparent concentrations of 117 those oxides without considering the complex stratigraphy of the objects’ surface. Those 118 results should be considered valid in a semi quantitative manner.” I understand that the results given in the respective table bring the group division identified by XRF analysis, however I believe that this manner of presenting the data about the colouring agents can lead to a misunderstanding. Since the data presented in a table or chart should be understandable without the need to read the text to comprehend it, I recommend the authors to reformulate this table considering the colours and its estimations in order to give the information in a clear way.
# 12 – Page 14, line 12 – Please correct Bertran et al., 2021 to Beltran et al., 2021.
# 13 – I recommend the authors reading more actual references to validate the discussions of the research findings, such as Bernard Gratuze, Teresa Medici and Inês Coutinho. Those references have a consistent work in the field of archaeometry and archaeological glass objects which may be helpful for the authors’ discussions.
#14 – Reference “Jackson, C. M., Greenfield, D. and Howie, L. A., 2012. An assessment of compositional and morphological changes in model archaeological glasses in an acid burial matrix. Archaeometry, Vol. 54, No. 3, pp. 489-507.” is duplicated. Please adjust it.
Comments on the Quality of English LanguageThe English needs minor revision.
Author Response
REVIEWER 1
The work “Investigating the Degradation of Mycenean Glass Artefacts Using Scientific Methods: Preliminary observations” aims to “offer strategies to tell apart degraded vitreous materials through characterizing glass and understanding its corrosion products with the application of non-invasive spectroscopic techniques- in particular X-ray based techniques and brings interesting data towards the relationship between the compositional changes of corroded vitreous artefacts and its visual effects. However, major alterations must be provided in order to guarantee the organization of the data as well as the better understanding of the work.
#1 – The abstract should briefly indicate the main results of the work. Please, add a synthesis of them in this section.
Reply to the reviewer
Thank you for your comment, the abstract has been enriched.
#2 – Page 1 “The main aim of this paper is to offer strategies to tell apart degraded vitreous materials through characterizing glass and understanding its corrosion products with the application of non-invasive spectroscopic techniques- in particular X-ray based techniques. Since the authors perform only XRF to characterize the studied samples, I recommend using “technique” instead of “techniques”.
Reply to the reviewer
Thank you, it has been adjusted.
# 3 – Despite affirming that the paper will not cover glass corrosion and its mechanisms, the authors describe in detail the main visual forms of alteration which occurs in archaeological glass. In my opinion, the description given should be summarized, so the author can explain better what the real issue of the work is. Otherwise, the audience may think the work .
Reply to the reviewer
Thank you for your comment. We feel that it is important for the readers to have an idea over what the terms we use to describe the alteration effects throughout the paper refer to. We did provide a more concise version though.
# 4 – Pg 2 – Lines 6-9 “In this paper we present the preliminary results obtained by LED Microscopy and micro-XRF applied on a subset of 12 objects of the overall assemblage of vitreous artefacts (126 artifacts). I suggest moving this information for section Materials and methods.
Reply to the reviewer
We did, thank you.
# 5 – Pg 2 – Lines 41-44 “Therefore, Mycenaean glasses, having as an alkali flux plant ashes and, thus, higher concentrations of potassium and given their small size are particularly vulnerable.”
Reply to the reviewer
There is no comment on that, but in case you wanted reference, we provided one.
# 6 – The paper does not present a Material and methods section, although it is a mandatory item for Heritage Manuscript Submissions. What were the techniques employed in this work? What were the conditions of the analysis performed by the authors? The information about the studied samples and the techniques are spread throughout the text and it must be organized in just one section. This description is fundamental to highlight the importance of the work and also to guarantee the reproducibility of the methodology.
Reply to the reviewer
We created a Materials and Methods section and gathered the data there. Thank you.
# 7 – The section of “results” is where the author presents the data gathered by the work, and not the conditions of the analysis. “Vitreous materials under study” (page 4), the first period of LED Microscopy subsection (page 5), initial paragraphs of Micro-XRF subsection, must be relocated to Materials and Methods section.
Reply to the reviewer
Thank you for the comment, we changed it.
# 8 – How can the authors explain the increase or depletion on the levels of chemical elements if only the altered layers were analysed?
Reply to the reviewer
In order to study the change in the chemical layers original material was required. This is why we chose the better preserved alternatives of each category. Relying on the ones that preserve a glassy layer we assess the difference from the corroded ones.
# 9 – I suggest the author to use “glassy layer” to describe the effect observed macro and microscopically, since “glassy phase” can induce the audience to think that it is related to chemical alteration identified by other type of analysis.
Reply to the reviewer
Thank you, it has been corrected.
# 10 – A scale should be included in the images of the samples and its micrography.
Reply to the reviewer
Another photo of the objects has been inserted in the text including a scale. The images from the Dynolight include a scale. The photos from the micro- XRF camera have been included only to show the exact area of analysis.
# 11 – Page 13, Lines 21-23 “The concentrations in Table 3 represent apparent concentrations of 117 those oxides without considering the complex stratigraphy of the objects’ surface. Those 118 results should be considered valid in a semi quantitative manner.” I understand that the results given in the respective table bring the group division identified by XRF analysis, however I believe that this manner of presenting the data about the colouring agents can lead to a misunderstanding. Since the data presented in a table or chart should be understandable without the need to read the text to comprehend it, I recommend the authors to reformulate this table considering the colours and its estimations in order to give the information in a clear way.
Reply to the reviewer
Unfortunately, we cannot understand this comment since the sentence “The concentrations in Table 3 represent apparent concentrations of 117 those oxides without considering the complex stratigraphy of the objects’ surface. Those 118 results should be considered valid in a semi quantitative manner.” Does not exist in the original submitted manuscript. Table 3 shows the composition of the chromophores in the specific objects.
# 12 – Page 14, line 12 – Please correct Bertran et al., 2021 to Beltran et al., 2021.
Reply to the reviewer
Thank you, it is now corrected.
# 13 – I recommend the authors reading more actual references to validate the discussions of the research findings, such as Bernard Gratuze, Teresa Medici and Inês Coutinho. Those references have a consistent work in the field of archaeometry and archaeological glass objects which may be helpful for the authors’ discussions.
Reply to the reviewer
Thank you, we did and were diverted through their work to works of other colleagues who have studied corrosion in LBA soda lime silica glasses.
#14 – Reference “Jackson, C. M., Greenfield, D. and Howie, L. A., 2012. An assessment of compositional and morphological changes in model archaeological glasses in an acid burial matrix. Archaeometry, Vol. 54, No. 3, pp. 489-507.” is duplicated. Please adjust it.
Reply to the reviewer
This has been changed.
Reviewer 2 Report
Comments and Suggestions for Authors
The paper “Investigating the Degradation of Mycenaean Glass Artefacts Using Scientific Methods: Preliminary observations.” investigates the compositional changes and the corresponding changes in appearance of glass artifacts. The authors started with a very long description of possible effects of degradation in ancient archeological glass followed by an evaluation of the samples using a Dynolight microscope.
Then, XRF analyses are introduced, and they used two standard glasses for calibration (Corning A and BAM005 in the text, BAM S005 in caption of table 2 while I found that only BAM S005A or B exist) without explaining how this calibration is achieved.
BAM S005 disappears in the rest of the text and nothing, referring to its use, is mentioned in all the paper while only Corning A seems to be used.
Corning A glass has a composition that approximate those of major antique glass types. So, the ratio between cps of each element in the samples and in the reference glass is surely an approximation of the real situation since no info are available about the real composition of undegraded glass. However, this could be acceptable when dealing with base glass.
But, when dealing with chromophores that can be present in the glass in highly variable amounts, I hardly understand how you can deduct the loss of Co, Cu or Fe from the comparison with Corning A standard.
How they got the data in table 3?
Finally, conclusion are a simple summary of what previously stated in the text and a repetition of information available in literature.
Author Response
REVIEWER 2
The paper “Investigating the Degradation of Mycenaean Glass Artefacts Using Scientific Methods: Preliminary observations.” investigates the compositional changes and the corresponding changes in appearance of glass artifacts. The authors started with a very long description of possible effects of degradation in ancient archeological glass followed by an evaluation of the samples using a Dynolight microscope.
Then, XRF analyses are introduced, and they used two standard glasses for calibration (Corning A and BAM005 in the text, BAM S005 in caption of table 2 while I found that only BAM S005A or B exist) without explaining how this calibration is achieved.
BAM S005 disappears in the rest of the text and nothing, referring to its use, is mentioned in all the paper while only Corning A seems to be used.
Corning A glass has a composition that approximate those of major antique glass types. So, the ratio between cps of each element in the samples and in the reference glass is surely an approximation of the real situation since no info are available about the real composition of undegraded glass. However, this could be acceptable when dealing with base glass.
But, when dealing with chromophores that can be present in the glass in highly variable amounts, I hardly understand how you can deduct the loss of Co, Cu or Fe from the comparison with Corning A standard.
Response to reviewer
ΒΑΜ005 is a typo and has been corrected. To tackle this “problem” you mentioned we changed the Corn A composition and we added mean values of typical LBA glass from already published works so as to make our comparison. In this sense it seems that we have the same behavior and therefore we keep the discussion as it is.
How they got the data in table 3?
Reply to the reviewer
These are the results of micro- XRF analysis.
Finally, conclusion are a simple summary of what previously stated in the text and a repetition of information available in literature.
Response to reviewer
Thank you for your comment. Conclusions have been enriched and refined.
Reviewer 3 Report
Comments and Suggestions for Authors
This study aims to shed light on the degradation mechanisms of vitreous artefacts and their implications for the archaeological record, which is a topic largely unexplored. The study focuses on deciphering the nature of corroded objects through detailed analysis can aid in the proper classification, management, and preservation of vitreous artefacts in the future. Employing XRF and LED techniques, the study aimed to characterize the glass and corrosion products of the Mycenaean glass artefacts, thereby contributing to a better understanding of the degradation mechanisms affecting these vitreous materials. By improving our understanding of the degradation mechanisms affecting these artefacts, improved strategies to preserve and interpret vitreous materials in archaeological contexts can be proposed, ultimately enhancing our knowledge of ancient glassmaking techniques and material culture.
Methodology is solid, although it lacks of some further explanations and rearrangements of tables and figures. Before publication, there are some changes that need to be addressed.
Figures lack of metric scale. Figure 1, for instance, show the actual 12 glassy samples but their size is uncertain. Similar situation faces Table 1, showing micro-XRF analysis areas and photos with no visible metric scale or indication of magnification, size, in table caption.
Table 1 may include information about the type of glass that is described in Figure 1 caption.
Line 58: what does this mean: “lime content is quite normal”? Please, be specific and provide justification.
Table 3. sigma refers to statistical precision of the measurement. How many measurements were done, 18 as well? Perhaps this value refers to relative standard deviation (RSD %) instead to standard deviation, which is the usual meaning for sigma?
Authors must provide a table with XRF chemical data. Discussion refers to amounts of different analytes, and the calculations of precision by analyzing corning A glass shows the chemical elements determined by this instrument. Surprisingly, there is no evidence of the actual data obtained on the glass samples. There is only disclosure of Fe, Co and Cu semiquantitative data, which is insufficient.
Finally, authors do not explain the archaeological background of their samples. Readers that are not familiar with Aegean Archaeology probably will get lost trying to understand the chronology of these glasses (not explained) or from which locations were retrieved. Additionally, authors do not comment on the actual chronology or chronologies of these samples. Are from different chronologies, centuries, or even typologies/classes or functionality? What we know as Mycenaean culture lasted for eight centuries and occupied a large portion of mainland Greece.
Author Response
REVIEWER 3
This study aims to shed light on the degradation mechanisms of vitreous artefacts and their implications for the archaeological record, which is a topic largely unexplored. The study focuses on deciphering the nature of corroded objects through detailed analysis can aid in the proper classification, management, and preservation of vitreous artefacts in the future. Employing XRF and LED techniques, the study aimed to characterize the glass and corrosion products of the Mycenaean glass artefacts, thereby contributing to a better understanding of the degradation mechanisms affecting these vitreous materials. By improving our understanding of the degradation mechanisms affecting these artefacts, improved strategies to preserve and interpret vitreous materials in archaeological contexts can be proposed, ultimately enhancing our knowledge of ancient glassmaking techniques and material culture.
Methodology is solid, although it lacks of some further explanations and rearrangements of tables and figures. Before publication, there are some changes that need to be addressed.
Figures lack of metric scale. Figure 1, for instance, show the actual 12 glassy samples but their size is uncertain. Similar situation faces Table 1, showing micro-XRF analysis areas and photos with no visible metric scale or indication of magnification, size, in table caption.
Reply to the reviewer
Another photo of the objects has been inserted in the text including a scale. The images from the Dynolight include a scale. The photos from the micro- XRF camera have been included only to show the exact area of analysis.
Line 58: what does this mean: “lime content is quite normal”? Please, be specific and provide justification.
Reply to the reviewer
Thank you, it has been specified in the text.
Table 3. sigma refers to statistical precision of the measurement. How many measurements were done, 18 as well? Perhaps this value refers to relative standard deviation (RSD %) instead to standard deviation, which is the usual meaning for sigma?
Reply to the reviewer
It’s a calculation of sigma area x100/fit area and refers to the statistical error of the peak intensities derived from the cumulative (sum) spectrum of the 18 measurements. It is not RSD.
Authors must provide a table with XRF chemical data. Discussion refers to amounts of different analytes, and the calculations of precision by analyzing corning A glass shows the chemical elements determined by this instrument. Surprisingly, there is no evidence of the actual data obtained on the glass samples. There is only disclosure of Fe, Co and Cu semiquantitative data, which is insufficient.
Reply to the reviewer
It is generally accepted that XRF analysis provides accurate quantitative data when certain requirements are fulfilled; for example, with respect to the quality and optimization of the experimental conditions ensuring adequate statistical precision, analytical range and sensitivity, if validated calibration and quantification procedures are applied using appropriate certified reference materials or if the analyzed sample presents homogeneous distribution of elements from the surface towards the depth. However, it is well known that the corroded glass presents an altered composition in the near-surface layers and depending on the type of corrosion certain elements are either enriched or depleted from the surface and likely follow a gradient like in-depth distribution. Any attempt to provide quantitative data in such a rather multi-layered sample system will generate effective and certainly non representative elemental concentrations. Due to the basic principles of X-rays interactions with matter, the elemental abundances that appear closer to the surface will be accounted with increased weight and those at deeper layers with much lower weight. The XRF quantification problem remains quite challenging for such archaeological materials and the use of supporting complementary techniques that can resolve the surface elemental stratigraphy (Confocal μ-XRF, Rutherford Back Scattering Spectrometry) is certainly needed.
Finally, authors do not explain the archaeological background of their samples. Readers that are not familiar with Aegean Archaeology probably will get lost trying to understand the chronology of these glasses (not explained) or from which locations were retrieved. Additionally, authors do not comment on the actual chronology or chronologies of these samples. Are from different chronologies, centuries, or even typologies/classes or functionality? What we know as Mycenaean culture lasted for eight centuries and occupied a large portion of mainland Greece.
Reply to the reviewer
Thank you for your comment, dating, a map and archaeological data based on the scope of this paper have been added.
Round 2
Reviewer 1 Report
Comments and Suggestions for Authors
I would like to thank you the author for providing all the information required.
Reviewer 3 Report
Comments and Suggestions for Authors
changes implemented by authors have improved the final manuscript.